# Role of *HNFA1* Gene Variants in Pancreatic Beta Cells Function and Glycaemic Control in Young Individuals with Type 1 Diabetes

**DOI:** 10.3390/biomedicines11071951

**Published:** 2023-07-10

**Authors:** Antonietta Robino, Gianluca Tornese, Davide Tinti, Klemen Dovc, Valeria Castorani, Andrea Conti, Roberto Franceschi, Ivana Rabbone, Riccardo Bonfanti, Tadej Battelino, Eulalia Catamo

**Affiliations:** 1Institute for Maternal and Child Health—IRCCS Burlo Garofolo, 34137 Trieste, Italy; antonietta.robino@burlo.trieste.it (A.R.); andrea.conti@burlo.trieste.it (A.C.); eulalia.catamo@burlo.trieste.it (E.C.); 2Center for Pediatric Diabetology, A.O.U. Città della Salute e della Scienza, 10126 Torino, Italy; davide.tinti@unito.it; 3Faculty of Medicine, University of Ljubljana, 1000 Ljubljana, Slovenia; klemen.dovc@mf.uni-lj.si (K.D.); tadej.battelino@mf.uni-lj.si (T.B.); 4Department of Endocrinology, Diabetes and Metabolism, University Children’s Hospital, University Medical Centre Ljubljana, 1000 Ljubljana, Slovenia; 5Department of Pediatrics, IRCCS San Raffaele Hospital, Diabetes Research Institute, 20132 Milano, Italy; castoraniv@gmail.com (V.C.); bonfanti.riccardo@hsr.it (R.B.); 6Division of Pediatrics, S. Chiara General Hospital, APSS, 38122 Trento, Italy; roberto.franceschi@apss.tn.it; 7Division of Pediatrics, Department of Health Sciences, Università del Piemonte Orientale, 28100 Novara, Italy; ivana.rabbone@uniupo.it

**Keywords:** *HNFA1* gene, beta cell function, glycaemic control, HbA1c

## Abstract

The HNF1A transcription factor, implicated in the regulation of pancreatic beta cells, as well as in glucose and lipid metabolism, is responsible for type 3 maturity-onset diabetes of the young (MODY3). HNF1A is also involved in increased susceptibility to polygenic forms of diabetes, such as type 2 diabetes (T2D) and gestational diabetes (GD), while its possible role in type 1 diabetes (T1D) is not known. In this study, 277 children and adolescents with T1D and 140 healthy controls were recruited. The following SNPs in *HNF1A* gene were selected: rs1169286, rs1169288, rs7979478, and rs2259816. Through linear or logistic regression analysis, we analyzed their association with T1D susceptibility and related clinical traits, such as insulin dose-adjusted glycated hemoglobin A1c (IDAA1c) and glycated hemoglobin (HbA1c). We found that rs1169286 was associated with IDAA1c and HbA1c values (*p*-value = 0.0027 and *p*-value = 0.0075, respectively), while rs1169288 was associated with IDAA1c (*p*-value = 0.0081). No association between *HNF1A* SNPs and T1D development emerged. In conclusion, our findings suggest for the first time that *HNF1A* variants may be a risk factor for beta cell function and glycaemic control in T1D individuals.

## 1. Introduction

*HNF1A* gene underlies the development of one of the most common forms of monogenic maturity-onset diabetes of the young (MODY), known as MODY3, a type of autosomal dominant diabetes with early onset. This rare type of diabetes is characterized by impaired insulin secretion due to mutations in a transcription factor involved in beta cell differentiation and function [1,2].

The *HNF1A* gene was also involved in the development of multifactorial forms of diabetes, probably affecting beta cell function. Variants in this gene were reported to contribute, for example, to increased susceptibility to type 2 Diabetes (T2D) [3,4] and gestational diabetes (GD) [5,6,7].

Generally, different *HNF1A* mutations lead to MODY3 diabetes than those involved in T2D and GD susceptibility. In MODY3, *HNF1A* Single Nucleotide Polymorphism (SNPs) act in the development of the disease already in heterozygosis by the need for sufficient levels of protein with a dominant negative effect of the mutant allele [8,9], whereas, in the T2D and GD susceptibility, *HNF1A* SNPs act in homozygosity, only partially affecting HNF1A function [10].

T1D is a polygenic form of diabetes, an autoimmune multifactorial disorder with impaired insulin production due to pancreatic beta cell destruction [11]. To date, there are numerous genes known to be involved in the predisposition to the development of T1D [12], but no effect of the *HNFA1* gene is known. Moreover, very few studies have already reported a possible role in T1D of other genes involved in the development of rarest forms of MODY, namely *NeuroD1* (MODY6), *ABCC8* (MODY12), and *KNCJ11*(MODY13) genes [13,14,15]. In addition to T1D susceptibility, Blasetti et al. [15] also found an association between rs5210 SNP in the *KCNJ11* gene and clinical features of T1D, such as BMI at onset and insulin requirement. The authors also observed higher C-peptide at onset, suggesting a degree of insulin resistance in rs5210 carrier and speculating on the similarity between T1D and T2D. To our knowledge, no other studies on the possible influence of MODY genes on clinical outcomes of T1D have been conducted. Nevertheless, the known impact of *HNFA1* mutations on pancreatic beta cells makes it reasonable to presume that polymorphisms in this gene may also be associated with key components in T1D pathogenesis, such as hyperglycaemia and beta cell function.

Therefore, in the present work, we investigated the association of common genetic variants in the *HNF1A* gene in young T1D individuals with T1D susceptibility and clinical traits related to beta cell function and glycaemic control. Specifically, we used HbA1c as a measure of glycaemic control, while IDAA1c was used as a marker of beta cell residual function [16,17,18].

## 2. Materials and Methods

### 2.1. Participants

For this study, we recruited 277 T1D individuals at Diabetes Units of IRCCS Burlo Garofolo (Trieste, Italy), Regina Margherita Children’s Hospital (Torino, Italy), University Medical Center (Ljubljana, Slovenia), IRCCS San Raffaele (Milano, Italy) and Santa Chiara Hospital (Trento, Italy). Inclusion criteria were diagnosis of T1D from at least 1 year, age between 6 and 21 years, and absence of other types of diabetes mellitus (i.e., type 2, monogenic diabetes, cystic fibrosis-related diabetes) [19].

We also enrolled 140 healthy controls (HC) from emergency departments. We excluded individuals with T1D or T2D diabetes (or other diabetes forms), obesity and other metabolic disorders, HbA1c > 6% (>42 mmol/mol), and family history of diabetes.

The ethics committee approved the protocol (CEUR-2018-Em-323-Burlo, KME-0120-65/2019/4). All participants and their parents (for participants aged <18 years) gave written informed consent before the enrolment.

### 2.2. Measurements and Protocol

For all participants, we collected demographic and anthropometric information, such as age, gender, height, and weight [20]. Standard deviation scores of BMI (BMI SDS) were calculated according to WHO reference charts [21] using the Growth Calculator 4 software, distributed by Italian Society of Pediatric Endocrinology and Diabetology (SIEDP, Bologna, Italy) (V0011, http://www.weboriented.it/gh4/, accessed on 27 August 2017). Moreover, for T1D participants, medical history (i.e., age at diagnosis, disease duration) and clinical information (i.e., insulin requirement, HbA1c) were collected. HbA1c was measured from finger pricks using portable instrumentation at outpatient clinics (DCA 2000 Analyzer System, Siemens, Munich, Germany).

According to HbA1c values, T1D participants were classified into two groups: HbA1c values < 7% (<53 mmol/mol) and HbA1c ≥ 7% (≥53 mmol/mol) [22].

IDAA1c was calculated as [HbA1c (%) + 4x insulin dose (units per kilogram per 24 h)] [23]. Then, participants were subdivided into two groups using IDAA1c ≤ 9 as a cut-off, previously associated with better pancreatic beta cell reserve [23].

### 2.3. DNA Extraction, Genotyping, and SNPs Selection

For each sample, DNA was extracted from saliva using the EZ1 DNA investigator kit (Qiagen, Milan, Italy) following the manufacturer’s protocols and then stored at −20 °C before analysis.

Genotyping was conducted by Illumina Infinium Global Screening Array (GSA v3.0).

Genotype calling was performed with the GenomeStudio software (V2.0, Illumina, Inc., San Diego, CA, USA). During the quality control (QC) step, we excluded the following: (1) samples with call rate < 95%, sex discrepancy, heterozygosity outside 6 standard deviations (sd) from the mean, identity by descent (IBD) proportion > 0.4; (2) duplicate SNPs, SNPs with missing call rate > 1% or with Hardy–Weinberg equilibrium (HWE) *p*-value < 1 × 10^−6^ [24].

For this study, we selected *HNF1A* SNPs in 12q23.31 location (GRCh37.p13 121416346-121440315). Monomorphic SNPs and SNPs with Minor Allele Frequency (MAF) < 5% were not considered. Linkage disequilibrium was calculated using PLINK software V1.9 (http://pngu.mgh.harvard.edu/purcell/plink/, accessed on 27 August 2017), and only the following SNPs with r^2^ > 0.85 were selected: rs1169288, rs1169286, rs7979478, and rs2259816 (Table 1).

### 2.4. Data Analysis

Participant characteristics were represented through percentages, means, and standard deviation (SD). Differences among HC and T1D individuals were analyzed by chi-squared tests to compare categorical data and *t*-tests to compare the means.

Association between diabetes susceptibility and *HNF1A* SNPs was analyzed by logistic regression analyses, gender and age-adjusted, while association between related-clinical traits and *HNF1A* SNPs was analyzed by logistic or linear regression analyses depending on whether the dependent variable is categorical or continuous. In all models, gender, age, and disease duration were included as covariates.

Statistical significance was set at *p*-value < 0.012, following Bonferroni correction (0.05/selected SNPs). All statistical analyses were performed with R software (V4.2.2., www.r-project.org, accessed on 31 October 2022).

## 3. Results

In this study, we enrolled 140 HC and 277 participants with T1D; 57% of HC and 46% of T1D were females (*p*-value = 0.044). The mean age was 12.5 ± 3.5 in HC and 13.2 ± 3.2 in T1D (*p*-value = 0.055). BMI SDS was higher in T1D individuals compared to HC (*p*-value =< 0.0001). Characteristics of all participants and additional clinical characteristics of T1D individuals are shown in Table 2.

In our sample, by comparing *HNF1A* SNPs frequencies between HC and T1D individuals, we did not observe statistically significant differences.

When we evaluated *HNF1A* polymorphisms and related clinical traits among T1D participants, we found an association of IDAA1c with rs1169286 T > C and rs1169288 A > C (*p*-value = 0.0027, beta = −0.39; *p*-value = 0.008, beta = −0.35, respectively). Specifically, TT carriers for rs1169286 and AA carriers for rs1169288 showed higher IDAA1c values (Figure 1).

rs1169286 SNP was also significantly associated with HbA1c values (*p*-value = 0.0075, beta = −0.23); specifically, higher HbA1c values were found in TT carriers. Although also for rs1169288 SNP AA carriers showed higher HbA1c values, the association did not reach the statistical significance after Bonferroni’s correction (*p*-value = 0.03) (Figure 2).

We also compared the distribution of rs1169286 and rs1169288 SNPs in T1D individuals classified according to IDAA1c and HbA1c values. For rs1169286, we found that the TT genotype was less frequent in T1D individuals with IDAA1c ≤ 9 and HbA1c < 7% (*p*-value = 0.0097 and *p*-value = 0.0052, respectively) (Table 3). The association of rs1169288 SNP with IDAA1c and HbA1c did not reach statistical significance.

## 4. Discussion

In this work, we analyzed the possible association of *HNF1A* SNPs with T1D and related clinical traits for the first time. Although *HNF1A* SNPs were not associated with T1D susceptibility, we detected an association of rs1169286 and rs1169288 SNPs with IDAA1c and HbA1c levels. More specifically, our results showed higher IDAA1c and HbA1c levels in TT individuals for the rs1169286 SNP. Moreover, higher IDAA1c was also found in AA carriers for rs1169288.

rs1169288 and other SNPs in the *HNF1A* gene have been previously associated with susceptibility to T2D and gestational diabetes [3,4,5,6,7], while, to our knowledge, no studies have already reported an association of rs1169286 with any polygenic form of diabetes, including T1D. Chiu et al. [25] reported an association of rs1169286 with insulin response in healthy normal-weight individuals, suggesting that this variant is an independent determinant of beta cell function and that may play a role in the pathogenesis of diabetes [25].

Mutations in the *HNF1A* gene are responsible for MODY3, characterized by severe pancreatic beta cell insulin secretory defects [26], although the clinical expression may vary considerably, and MODY3 subjects may present with a defect in insulin secretion, as well as the full spectrum of complications typical of diabetes, such as microvascular complications or those involving the kidneys [26,27,28].

The function of HNF1A is well known and supports its involvement in beta cell function. HNF1A is expressed in several human tissues, including liver and pancreas tissues, which controls the transcriptional expression of many genes playing varied and important roles [29]. For example, in the liver, HNF1A regulates genes contributing to the metabolism of substances such as glucose and fat [30,31]. In the pancreas, HNF1A controls genes involved in beta cell maturation and growth, as well as insulin secretion [32]. *HNF1A* mutations may result, for example, in reducing insulin secretion by binding directly to the promoter region of the insulin gene and positively regulating its activation [33,34]. *HNF1A* variations may also change the expression of enzymes involved in mitochondrial glucose metabolism [35], representing, in conclusion, a very important transcription factor for the maintenance of beta cell function. Based on this evidence, we can speculate that similar mechanisms may be responsible for altered beta cell function and glycaemic control in T1D.

In the present study, we also observed that the TT genotype for rs1169286 was less frequent in T1D individuals with IDAA1c ≤ 9, supporting a potential effect of this SNP on beta cell function. Therefore, we can hypothesize that better IDAA1c values, a marker of beta cell residual function [16,17,18], in CC carriers for *HNF1A* SNPs may be linked to a possible higher residual beta cell activity.

Moreover, IDAA1c ≤ 9 has been associated in T1D individuals with a lower frequency of microvascular complications [36], justifying our finding on the low frequency of TT genotype for rs1169286 also in the HbA1c < 7% participants.

Our results on the association between rs1169286 SNP and HbA1c suggest a possible effect of an *HNF1A* gene on glycaemic control, which is one of the most important determinants in the development of microvascular complications. In this light, studies on the development of cardiovascular diseases over time in the different *HNF1A* genotype carriers would be of interest.

In contrast to previous studies reporting an association of MODY genes and T1D susceptibility [13,14,15], in the present work, we did not find significant differences by comparing *HNF1A* SNPs frequencies between HC and T1D individuals.

The findings of this study are subject to some limitations. For example, CGM (Continuous Glucose Monitoring) data or C-peptide values (as a measure of beta cell function) are not available for this study; however, we analyzed IDAA1c, which was reported as an easy and fast alternative to evaluate pancreatic beta cell function [16,17,18]. The limited number of variants in the *HNF1A* locus included in our chip array may be another limit and did not allow us to analyze rare variants that, for example, were previously implicated in T2D susceptibility [37]. The presence of sex and standardized BMI differences between T1D and HC may represent an additional limitation.

Finally, our results should be carefully interpreted in light of the small sample size that may underlie the lack of association between T1D and *HNF1A* SNPs. Despite these limitations, our results suggest, for the first time, that *HNF1A* variants may be a risk factor for beta cell function and glycaemic control and may be useful in the early identification of individuals with T1D who could benefit from early and more focused attention on prevention of T1D-associated complications.

Additional investigations are needed to confirm our findings and to better understand the exact mechanism by which the *HNF1A* gene may affect T1D clinical outcomes.

## Figures and Tables

**Figure 1 biomedicines-11-01951-f001:**
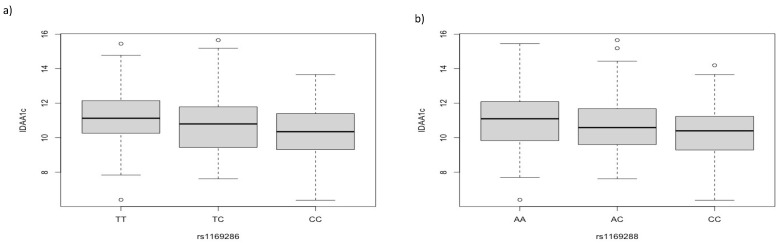
IDAA1c values by *HNFA1* SNPs. (**a**) Boxplot showing higher IDAA1c values in TT carriers for rs1169286 (*p*-value = 0.0027); (**b**) boxplot showing higher IDAA1c values in AA carriers for rs1169288 (*p*-value = 0.008).

**Figure 2 biomedicines-11-01951-f002:**
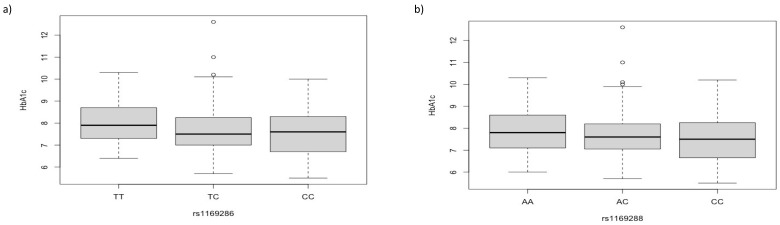
HbA1c values by *HNF1A* rs1169286 SNP. (**a**) Boxplot showing higher HbA1c values in TT carriers for rs1169286 (*p*-value = 0.0077); (**b**) boxplot showing HbA1c values by rs1169288 (*p*-value = 0.03, not statistically significant).

**Table 1 biomedicines-11-01951-t001:** *HNF1A* Single Nucleotide Polymorphisms.

*HNF1A* SNPs	Type	Nucleotide Change (Aminoacid Change)	Chr:Position	Reference Allele	Alternative Allele	MAF
rs1169288	Missense Variant	c.79A > C (p.Ile27Leu)	12:121416650	A	C	36.0%
rs1169286	Intron	c.326 + 2159T > C	12:121419056	T	C	45.0%
rs7979478	Intron	c.326 + 3366A > C	12:121420263	A	G	56.5%
rs2259816	Synonymous Variant	c.1620G > A (p.Val540=)	12:121435587	G	T	41.0%

Chr = Chromosome; MAF = Minor Allele Frequency.

**Table 2 biomedicines-11-01951-t002:** Healthy Controls (HC) and type 1 diabetes (T1D) participants’ characteristics.

	HC (*n* = 140)	T1D (*n* = 277)	*p*-Value
Sex, (%, females)	57%	46%	0.044
Age (years, mean ± SD)	12.5 ± 3.5	13.2 ± 3.2	0.055
Standardized BMI (mean ± SD)	−0.39 ± 1.1	0.16 ± 1.1	<0.0001
Disease duration (years, mean ± SD)	-	5.4 ± 3.6	
HbA1c (%, mean ± SD)	-	7.8 ± 1.0	
IDAA1c (U/kg/die, mean ± SD)	-	10.8 ± 1.6	

Differences were computed by *t*-test and chi-square test, as appropriate.

**Table 3 biomedicines-11-01951-t003:** Distribution of rs1169286 and rs1169288 genotypes in T1D participants classified by IDAA1c and HbA1c values.

Clinical Parameters	rs1169286			*p*-Value	rs1169288			*p*-Value
IDAA1c	TT	TC	CC	**0.0097**	AA	AC	CC	0.044
≤9 (*n* = 39)	15.0%	54.0%	31.0%		31.0%	49.0%	20.0%	
>9 (*n* = 238)	33.0%	4.0%	1.0%		43.0%	44.0%	13.0%	
HbA1c				**0.0052**				0.07
<7 (*n* = 64)	19.0%	51.0%	30.0%		37.5%	40.5%	22.0%	
≥7 (*n* = 213)	34.0%	48.0%	18.0%		43.0%	45.0%	12.0%	

Significant results were indicated in bold (*p* < 0.012). Differences were computed by logistic regression models using sex, age, and disease duration as covariates.

## Data Availability

The data presented in this study are available on request from the corresponding author. They are not publicly available due to privacy reasons.

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
