# Peer review of "Role of HNFA1 Gene Variants in Pancreatic Beta Cells Function and Glycaemic Control in Young Individuals with Type 1 Diabetes"

_biomedicines, 2023, doi:10.3390/biomedicines11071951_

Round 1

Reviewer 1 Report

Title: Role of HNFA1 gene variants in pancreatic beta cells function and glycemic control in young individuals with Type 1 diabetes

Authors: Antonietta Robino, Gianluca Tornese, Davide Tinti, Klemen Dovc, Valeria Castorani, Andrea Conti, Roberto Franceschi, Ivana Rabbone, Riccardo Bonfanti, Tadej Battelino and Eulalia Catamo

General Comment:

Type 1 diabetes (T1D) is a complex multifactorial disease, which development is determined by numerous genetic and environmental factors. The genetic background of T1D is heterogeneous, which influences the varied course of the disease, including difficulties with glycaemic control and the development of chronic complications.  In their study, Antonietta Robino et al analyzed the effect of variants in the HNFA1 gene on pancreatic beta cell function and glycemic control in young individuals with T1D. They found that although the frequency of these variants in diabetic patients does not differ from that observed in healthy controls, HNFA1 polymorphisms may determine beta cell function and thus glycaemic control. The research hypothesis is well justified, the methodology adequate, the results clearly presented and the discussion extensive, so I have only 2 minor comments that the authors should consider before the manuscript is accepted for publication.

Minor revisions:

Material and methods:

1)      Please add information on whether a power analysis was carried out - was the number of participants in the study sufficient to observe true correlations? An insufficient number of subjects may be the reason for the lack of differences in genotype distribution between T1D patients and controls. This is worth mentioning in the discussion when listing potential limitations of the study.

Discussion

2)      The discussion is very comprehensive and comments on the potential impact of the studied variants on the course of T1D. However, it is worth commenting on the fact that the same variants occur with similar frequency in healthy controls and do not determine glycemia.

Author Response

General Comment:

Type 1 diabetes (T1D) is a complex multifactorial disease, which development is determined by numerous genetic and environmental factors. The genetic background of T1D is heterogeneous, which influences the varied course of the disease, including difficulties with glycaemic control and the development of chronic complications.  In their study, Antonietta Robino et al analyzed the effect of variants in the HNFA1 gene on pancreatic beta cell function and glycemic control in young individuals with T1D. They found that although the frequency of these variants in diabetic patients does not differ from that observed in healthy controls, HNFA1 polymorphisms may determine beta cell function and thus glycaemic control. The research hypothesis is well justified, the methodology adequate, the results clearly presented and the discussion extensive, so I have only 2 minor comments that the authors should consider before the manuscript is accepted for publication.

Minor revisions:

Material and methods:

  • Please add information on whether a power analysis was carried out - was the number of participants in the study sufficient to observe true correlations? An insufficient number of subjects may be the reason for the lack of differences in genotype distribution between T1D patients and controls. This is worth mentioning in the discussion when listing potential limitations of the study.

We modified limitation section accordingly to the comments of the two reviewers and we added the small sample size as limitation. We did not perform a priori sample calculation size. However, to our knowledge other studies on MODY SNPs and T1D (below reported) included smaller sample size:

Soltani Asl S, Azimnasab-Sorkhabi P, Abolfathi AA, Aghdam YH. Identification of nucleotide polymorphism within the NeuroD1 candidate gene and its association with type 1 diabetes susceptibility in Iranian people by polymerase chain reaction-restriction fragment length polymorphism. J Pediatr Endocrinol Metab 2020;33(10):1293-1297. doi: 10.1515/jpem-2019-0441.

Reddy S, Maddhuri S, Nallari P, Ananthapur V, Kalyani S, Murali K, Nirmala. Association of ABCC8 and KCNJ11 gene variants with type 1 diabetes in south Indians. Egyptian Journal of Medical Human Genetics 2021;22:27. https://doi.org/10.1186/s43042-021-00149-w.

Blasetti A, Castorani V, Comegna L, Franchini S, Prezioso G, Provenzano M, et al. Role of the KCNJ gene variants in the clinical outcome of type 1 diabetes. Horm Metab Res. 2020;52(12):856-860. doi: 10.1055/a-1204-5443.

Discussion

2)      The discussion is very comprehensive and comments on the potential impact of the studied variants on the course of T1D. However, it is worth commenting on the fact that the same variants occur with similar frequency in healthy controls and do not determine glycemia.

As suggested by reviewer, we the following sentence: “In contrast to previous studies reporting an association of MODY genes and T1D susceptibility [13-15], in the present work we did not find significant differences by comparing HNF1A SNPs frequencies between HC and T1D individuals.”.

Reviewer 2 Report

The authors enrolled 277 children and adolescents with T1D and 140 healthy controls to evaluate the effect of HNF1A SNPs on HbA1c and other laboratory parameters. The following SNPs in HNF1A gene were selected: rs1169286, rs1169288, rs7979478, 22 rs2259816. Through linear or logistic regression analysis, they analyzed their association with T1D susceptibility and related-clinical traits, such as Insulin-dose adjusted glycated hemoglobin A1c (IDAA1c) and glycated hemoglobin (HbA1c). They found that rs1169286 was associated with IDAA1c and HbA1c values, while rs1169288 was associated with IDAA1c. No association between HNF1A SNPs and T1D development emerged. They concluded that HNF1A variants may be a risk factor for beta cell function and glycemic control in T1D individuals.

It is a well designed and nicely presented study.

Comments

1.      There as a significant difference in BMI values between controls and T1D patients. It should be mentioned as a limitation of the study.

2.      Fig.1. and Fig 2. p values are missing. Figure legends should be completed.

Author Response

The authors enrolled 277 children and adolescents with T1D and 140 healthy controls to evaluate the effect of HNF1A SNPs on HbA1c and other laboratory parameters. The following SNPs in HNF1A gene were selected: rs1169286, rs1169288, rs7979478, 22 rs2259816. Through linear or logistic regression analysis, they analyzed their association with T1D susceptibility and related-clinical traits, such as Insulin-dose adjusted glycated hemoglobin A1c (IDAA1c) and glycated hemoglobin (HbA1c). They found that rs1169286 was associated with IDAA1c and HbA1c values, while rs1169288 was associated with IDAA1c. No association between HNF1A SNPs and T1D development emerged. They concluded that HNF1A variants may be a risk factor for beta cell function and glycemic control in T1D individuals.

It is a well designed and nicely presented study. 

Comments

  1. There as a significant difference in BMI values between controls and T1D patients. It should be mentioned as a limitation of the study.

As suggested, we added the following sentence: “The presence of sex and standardized BMI differences between T1D and HC may represent an additional limitation.”  Overall, we modified limitation section accordingly to the comments of the two reviewers.

  1. Fig.1. and Fig 2. p values are missing. Figure legends should be completed.

We thank the reviewer for the comment. We modified figure legends and we added related p-values.